# pH Measurements Using Leaky Waveguides with Synthetic Hydrogel Films

**DOI:** 10.3390/mi16020216

**Published:** 2025-02-14

**Authors:** Victoria Wensley, Nicholas J. Goddard, Ruchi Gupta

**Affiliations:** 1School of Chemistry, University of Birmingham, Birmingham B15 2TT, UK; vic.wensley@protonmail.com; 2Process Instruments (UK) Ltd., Turf St, Burnley BB11 3BP, UK; nick.goddard@processinstruments.net

**Keywords:** leaky waveguides, hydrogels, pH

## Abstract

Leaky waveguides (LWs) are low-refractive-index films deposited on glass substrates. In these, light can travel in the film while leaking out at the film–substrate interface. The angle at which light can travel in the film is dependent on its refractive index and thickness, which can change with pH when the film is made of pH-responsive materials. Herein, we report an LW comprising a waveguide film made of a synthetic hydrogel containing the monomers acrylamide and N-[3-(dimethylamino)propyl]methacrylamide (DMA) and a bisacrylamide crosslinker for pH measurements between 4 and 8. The response of the LW pH sensor was reversible and the response times were 0.90 ± 0.14 and 2.38 ± 0.22 min when pH was changed from low to high and high to low, respectively. The reported LW pH sensor was largely insensitive to typical concentrations of common interferents, including sodium chloride, urea, aluminum sulfate, calcium chloride, and humic acid. Compared to a glass pH electrode, the measurement range is smaller but is close to the range required for monitoring the pH of drinking water. The pH resolution of the hydrogel sensor was ~0.004, compared to ~0.01 for the glass electrode.

## 1. Introduction

In the UK, the Water Supply (Water Quality) Regulations 2018 [1] specify that the pH of water supplied to consumers should be within the range of 6.5 to 9.5, which is in line with World Health Organization (WHO) recommendations [2]. Currently, pH is usually measured potentiometrically between a pH-sensitive glass electrode and a silver/silver chloride reference electrode [3]. Most pH electrodes are of the combination type, where the reference electrode is contained within the body of the glass electrode and contacts the solution being measured through a glass or ceramic frit. Glass electrodes can measure pH over a wide range, often from 0 to 14, with a resolution of ~0.01 [4]. This is is more than sufficient for water quality monitoring. However, they have the disadvantage of fragility of the glass membrane and slow response times in low-ionic-strength solutions. An additional disadvantage in waters containing organic matter is the clogging of the reference electrode frit, causing erroneous measurements.

Optical methods of measuring pH typically use colorimetric or fluorimetric pH indicator reagents [5,6]. Colorimetric methods can be either visual, using test strips or visual comparators, or photometric. These techniques are used to measure the absorbance of a known concentration of reagent added to a water sample. For water quality, phenol red is often used as an indicator [7] as it shows a pronounced color change from yellow to red over the pH range from 6.8 to 8.2. Fluorimetric indicators are less often used for water quality monitoring as the reagents are more expensive than colorimetric indicators. However, they can be used to measure pH in two ways—by measuring the intensity of fluorescence [5] and the fluorescence lifetime [8]. Optical pH measurement can be confounded by the presence of color or turbidity in the water sample [9]. Additionally, the indicators can be destroyed by the presence of certain elements, species such as chlorine added to water as a disinfectant.

In earlier work, the authors used a metal-clad leaky waveguide (MCLW) to perform broadband absorption spectroscopy in very small volumes (3.2 nL) in order to minimize reagent usage for pH measurements [10]. The MCLW was fabricated by the vacuum deposition of titanium followed by the spin-coating of an agarose film. Transmission grating at the output of the MCLW provided dispersion of the white light source and permitted the measurement of absorption over the wavelength range from 450 to 700 nm. The dye bromothymol blue was used as the colorimetric indicator at a concentration of 250 µM. This method significantly reduced the volume of sample and reagent required, but still had all the disadvantages of colorimetric pH methods.

To overcome the need to add a colorimetric indicator to the sample, in later work, the authors created a diffractive sensor where laser interference lithography was used to induce periodic variation in the concentration of the pH-sensitive dye fluorescein isothiocyanate (FITC), immobilized in an acrylamide hydrogel [11,12]. This formed an amplitude grating whose diffraction efficiency depended on the pH of the solution into which the grating was immersed. Unlike conventional absorbance measurement, which produces a sigmoidal change in absorbance with pH limiting the range to ±1 pH units around the pK_a_ of the indicator, the grating sensor was linear over a range of 4 pH units [11].

Another method of measuring pH involves using the swelling of hydrogels containing ionizable groups [13,14]. Hydrogels typically swell when ionizable groups are charged because of electrostatic repulsion. In principle, this is reversible and overcomes the need for indicators. Various transduction methods have been employed to convert the swelling and shrinking of pH-responsive hydrogels into a useful signal, including refractometry [15,16], holograms [17,18], Bragg gratings [19], conductimetry [20], quartz crystal microbalances [21,22], and micro cantilevers [23]. Some of these transducers are complex and expensive to fabricate, making them less suitable for mass adoption to replace glass pH electrodes. Hydrogel-based optical pH sensors can be broadly divided into three classes: (i) those based on refractive index changes caused by the swelling/shrinking of the hydrogel as a result of pH changes [15,16,24,25,26], (ii) those based on volume changes [13,27,28] and (iii) those using hydrogels to entrap or immobilize a pH indicator [29,30,31]. The latter class of sensor has all the drawbacks of indicators and is only briefly considered here. We can distinguish refractometric from volumetric sensors as refractometric sensors do not sense throughout the entire volume of the hydrogel, unlike volumetric sensors. Many of the refractometric sensors use evanescent wave techniques to probe the refractive index of a thin (<1 µm) layer of the hydrogel adjacent to the transducer surface.

To create a simple and easy-to-fabricate optical pH sensor, we recently described an LW-based pH sensor using stacked waveguide layers of chitosan (pH-sensitive) on top of agarose (pH-insensitive) [32]. The chitosan layer was doped with the dye reactive blue 4 (RB4) to induce a dip in reflectivity at the resonance angles of both the agarose and chitosan waveguides. The agarose layer acted as a reference for reducing common-mode effects such as temperature and sample refractive index changes by a factor of 7, while allowing pH measurements between 3 and 9.

The use of natural materials such as agarose and chitosan presents challenges in ensuring reproducible sensor fabrication as there are natural variations, even in different batches from the same manufacturer. Indeed, rigorous purification is required to ensure that chitosan in particular produces reproducible results [33]. To overcome many of the disadvantages of natural materials, the present work describes a pH sensor based on LWs fabricated from synthetic hydrogels. The resulting LW sensor comprises a thin layer of pH-responsive hydrogel on a glass substrate, requiring no complex fabrication and giving a usable response between pH 4 and 7. This type of sensor falls between purely refractometric and volumetric sensors, as the LW mode occupies the entire thickness of the hydrogel layer. As such, it senses the optical thickness of the hydrogel layer—the product of the physical thickness and refractive index. The sensor itself is very simple, consisting of a glass substrate with a cast hydrogel layer on top. No dye, fluorophore, nanoparticles, optical fibers, metal layers, or micro/nanostructures are required to make a working pH sensor, unlike most of the sensors discussed above.

This type of sensor is well suited to, for example, environmental monitoring as it could, in principle, be developed into a probe for insertion into pipelines or for dipping into watercourses. The simplicity of the sensor also means that it could be fabricated in volume at low cost. It could also be used in healthcare applications for ex vivo blood pH measurement as only tiny sample volumes (10–20 µL) are required.

## 2. Materials and Methods

### 2.1. Chemicals and Materials

We purchased 1.2–1.5 mm thick microscope glass slides (MIC2154) from Scientific Laboratory Supplies (Nottingham, UK). Acrylamide 40% (*w*/*v*), N, N-methylene bisacrylamide, ammonium persulfate (APS), N, N, N′, N′-tetramethylethylenediamine (TEMED), ethanol, polystyrene latex beads (1.1 µm mean particle size), mono-, di-and tri-basic sodium phosphate salts, 1 M sodium hydroxide, 1 M hydrochloric acid, sodium chloride, urea, aluminum sulfate, calcium chloride, and humic acid were procured from Sigma Aldrich (Gillingham, UK). Decon 90 and Cargille refractive index matching immersion oil type B were procured from Fisher Scientific (Loughborough, UK). N-[3-(dimethylamino)propyl]methacrylamide (DMA) was procured from ABCR (Karlsruhe, Germany). Chloro(dimethyl)vinylsilane (CDMVS) was purchased from TCI Chemicals (Oxford, UK). Nitrile O-rings were purchased from Ashton Seals (Barnsley, UK). Bootlace ferrules were bought from RS Components (Corby, UK).

### 2.2. Fabrication of LWs

LWs were fabricated by depositing hydrogel films on glass slides via casting. A schematic showing the procedure of casting is provided in Figure 1.

A 100 µL drop of hydrogel precursor solution was sandwiched between two glass slides, where the bottom glass slide was coated with CDMVS and the top glass slide was patterned with either photoresist spacers or spotted with polystyrene latex beads on all four corners. A brass weight of 500 g was placed on top of glass–precursor solution–glass sandwich, while the solution was polymerized and immersed in deionized water for at least 2 h before being gently pried apart. This resulted in a hydrogel film being deposited on the bottom glass slide because of the covalent linkages formed between CDMVS and the hydrogel during polymerization.

We obtained ~25 × 25 mm top and bottom glass squares used for casting by cutting microscope slides using a diamond scribe. Before use, the glass squares were sonicated for 30 min each in Decon 90 detergent solution, deionized water, and ethanol. Then, they were dried in air. To obtain bottom glass squares for casting, cleaned glass squares were immersed in a 0.2% (*v*:*v*) solution of CDMVS in toluene for 30 min, transferred to clean toluene for 5 min, and dried on soft tissue paper. To obtain top glass squares for casting, polystyrene latex beads were deposited on all four corners of clean glass slides by pipetting 1 µL of a 1% (*w*:*v*) suspension of beads in water and drying at 60 °C overnight. Alternatively, top glass squares were obtained for casting by spin coating Megaposit SPR220-3.0 photoresist (A-Gas Electronic Materials, Warwickshire, UK) at 7000 rpm for 30 s, exposing it to UV light through a photomask, and immersing it in MF-24A developer to obtain rectangular photoresist spacers on top glass slides, as shown in Figure 1. For beads, based on the size distribution given by the supplier, hydrogel layer thickness was between 1.0 and 1.2 µm. Based on the photoresist datasheet, the thickness of the photoresist spacers was 2.0 ± 0.01 µm.

The hydrogel precursor solution contained monomers acrylamide and DMA, as well as a bisacrylamide crosslinker. Additionally, the precursor solution contained 1.25 µL of TEMED and 12.5 µL of 10% (*w*:*v*) APS, prepared in deionized water, to initiate free radical polymerization. The total weight to volume ratio of monomers and crosslinker was either 4%, 5%, or 6%. The molar ratio of DMA to total monomer varied between 0% and 40%. The molar ratio of the crosslinker with respect to the monomers was 3%.

### 2.3. Instrumentation

A schematic of the LW instrument is shown in Figure 2, with details provided in previously published work [34].

Briefly, light was incident on the LW through the face of an equilateral prism (Qioptic Photonics, Denbighshire). A thin layer of refractive index matching oil was placed between the LW and the equilateral prism to enable good optical contact. Light of a 650 nm wavelength from a superluminescent diode (EXS210035-02, Exalos AG, Schlieren, Switzerland) was passed through an achromatic doublet lens to obtain a collimated beam with a 25 mm diameter. The collimated beam was passed through a polarizer to obtain a transverse electric (TE) polarized light. The light was then passed through a cylindrical lens to obtain a wedge-shaped beam, which was directed into the prism to illuminate the LW mounted on top of the prism. Light reflected from the LW was passed through a cylindrical lens and recorded on a camera (UI-1492LE, IDS Imaging, Obersulm, Germany) with a resolution of 10 mega pixels and 2748 × 3840 pixels, located on a pitch of 1.67 µm. The angular resolution of the instrumentation was 4 × 10^−4^ deg based on three times the standard deviation of the angular noise, or about 2.7 × 10^−6^ RIU based on the refractive index sensitivity of 148.7 deg RIU^−1^ measured using glycerol solutions.

Solutions were introduced on the LW using a single-channel flow cell, as shown in Figure 2. The flow cell was held in place using a metallic clamp, which was connected to a water bath to maintain the temperature of the assembly at 25 °C. The flow cell was made by machining a 10 mm long, 4 mm wide, and 0.2 mm deep cavity in a 3 mm thick black poly(methyl methacrylate) (PMMA). The cavity served as the fluidic channel. The cavity was surrounded by a groove 1 mm wide and 0.75 mm deep in order to mount a 1 mm thick O-ring. The O-ring allowed us to form a good seal between the flow cell and the LW, and hence liquids could be pumped into the fluidic channel without any leakage. Bootlace ferrules were glued in through holes at either end of the cavity in PMMA to form fluidic inlets and outlets. The inlet was connected to a Gilson F155001 minipuls 3 peristaltic pump (Scientific Laboratory Supplies, Nottingham, UK) using silicone tubing. A flow rate of 0.2 mL min^−1^ was used throughout. One end of another piece of silicone tubing was connected to the flow cell outlet and the other end of the silicone tubing was left in a large beaker to collect waste liquid.

### 2.4. Methodology

To study the pH response of LWs, 10 mM phosphate buffers of selected pH values were introduced into the flow cell and shifts in resonance angles were monitored in real time using an in-house program written in C++ Builder v12.2. A solution of selected pH values flowed until the shift in resonance angle reached a stable value, following which a solution with another pH value was flowed, and the process was repeated.

The in-house written program allowed users to make rectangles of selected widths and heights on a camera image. For each rectangular box, grayscale values were averaged over the width of the rectangle (i.e., distance along the width of the flow channel) and plotted as a function of the height of the rectangle (i.e., angle of incidence). Plots of grayscale values, as a function of incidence angle, resulted in a peak–dip pair at the resonance angle. The center of gravity algorithm was used to find the position of the dip, which was used to determine the resonance angle in real time.

## 3. Working Principle

The simplest LW is a few microns thick film and is made of a low-refractive-index material, such as a hydrogel, deposited on a glass substrate with a sample solution placed on top of the film. At appropriate combinations of geometrical and optical parameters, light can travel in the film but leaks out at the film–substrate interface. Hence, the low-refractive-index (hydrogel) film acts as a waveguide that is leaky.

The operation of the LW can be understood by considering that the refractive index of the waveguide film (*n_w_*) is lower than that of the substrate (*n_s_*) but higher than that of the sample solution (*n_c_*). The specific angles of incidence at which light can travel in waveguide films are called resonance angles (*θ_R,int_*). At a resonance angle, as shown in Figure 3, light travels in the waveguide film, bouncing between the waveguide–sample and waveguide–substrate interfaces before it comes out of the waveguide film. Light traveling in the waveguide film undergoes total internal reflection (TIR) at the waveguide–sample interface. However, at the waveguide–substrate interface, light undergoes the Fresnel reflection and hence is partially lost (shown by thin red arrow in Figure 3). Light cannot undergo TIR at the waveguide–substrate interface because *n_w_* < *n_s_*. This implies that, in contrast to conventional waveguides where light undergoes TIR at both interfaces, light undergoes TIR at one interface and a Fresnel reflection at the other interface. Hence, a small fraction of light leaks out from the waveguide–substrate interface in LWs. As a result, light can only travel from a few tens of microns to tens of millimeters in the hydrogel film of LWs. This travel distance is sufficient for sensing applications.

The mode equation (Equation (1)) for an LW can be generated by summing the phase shifts that light experiences as it travels through the waveguide (*Φ_z_*) and reflections at the waveguide–sample (*Φ_w,c_*) and waveguide–substrate (*Φ_w,s_*) interfaces, and then setting this to be equal to an integer multiple of 2π [35]. Equation (1) is the result, while Equations (2)–(4) give the expressions for each phase shift for TE polarized light.(1)$$\Phi_{tot}=2\Phi_{z}+\Phi_{w,s}+\Phi_{w,c}=2m\pi$$(2)$$\Phi_{z}=\frac{2\pi h}{\lambda }\sqrt{n_{w}^{2}-N^{2}}$$(3)$$\Phi_{w,c}=-2{tan}^{-1}\left[\sqrt{\frac{N^{2}-n_{c}^{2}}{n_{w}^{2}-N^{2}}}\right]$$(4)$$\Phi_{w,s}=\pi$$

By substituting Equations (2)–(4) into Equation (1) and rearranging them, we can obtain Equation (5), where *N* is the mode effective refractive index and must lie in between *n_c_* and *n_w_*, while *h* is the waveguide film thickness and *λ* the vacuum wavelength of light. Equation (5) shows that solutions of this equation are those for values of N that result in a phase shift that is an odd-integer multiple of π. These are the modes of LW, and the thicker the waveguide (larger *h*), the more modes that can propagate in the LW. Unfortunately, Equation (5) cannot be solved analytically and must be solved numerically or graphically to obtain values of N.(5)$$\left(2m-1\right)\pi =\frac{4\pi h}{\lambda }\sqrt{n_{w}^{2}-N^{2}}-2{tan}^{-1}\left[\sqrt{\frac{N^{2}-n_{c}^{2}}{n_{w}^{2}-N^{2}}}\right]$$

Once N has been determined, Equation (6) gives the resonance angles (*θ_R,int_*) marked in Figure 3. For a flat substrate, *θ_R,int_* is complex and hence physically unrealizable. Prism coupling, conducted using a suitable prism placed underneath the substrate, overcomes this limitation [35]. The external resonance angle *θ_R,ext_*, which is marked in Figure 3, is given by Equation (7), where *ϕ* is the angle between the substrate and the prism faces. In this work, an equilateral prism was used, and so *ϕ* was 60°.(6)$$\theta_{R,int}={sin}^{-1}\frac{N}{n_{s}}$$(7)$$\theta_{R,ext}=\phi +\sin^{-1}\left(N\cos\phi -\sqrt{n_{s}^{2}-N^{2}}\sin\phi \right)$$

In previous work, resonance angles were visualized in light reflected from LWs and captured on a camera by either depositing a titanium layer between glass and the waveguide film (i.e., MCLW) or by incorporating a dye into the waveguide film (i.e., DDLW). Both LW configurations have disadvantages: MCLWs require the vacuum deposition of the metal layer, while DDLWs take up some immobilization sites with dyes, which may also increase non-specific adsorption. For this work, a third method was employed using very low-refractive-index waveguide films, as described by the authors in 2020 [36,37]. For LWs with very low-refractive-index waveguide films, the phase shift at resonance varies rapidly by 2π with the angle of incidence. This results in Fresnel diffraction, giving rise to observable changes specifically, dip–peak pairs in the reflectivity of the LW, even without a metal layer or dye doping.

## 4. Results and Discussion

### 4.1. pH Response

Based on Equation (5), *N* depends on refractive index and thickness (i.e., *n_w_* and *h*, respectively) of waveguide films. This implies that resonance angle *θ_R,ext_* changes because of the swelling/shrinking of hydrogel films in response to pH. This implies that shifts in resonance angles of LWs can be used to measure pH. In this work, pH-sensitive waveguide film was synthesized from a mixture of the monomers acrylamide and N-[3-(dimethylamino)propyl]methacrylamide (DMA), with N, N-methylene bisacrylamide as the crosslinker, as described in Section 2.2. The chemical structures of the monomers and the crosslinker are shown in Figure 4, highlighting that DMA contained a tertiary amine, which can protonate/deprotonate depending on solution pH. When tertiary amines are protonated, because of electrostatic repulsion, DMA-containing hydrogels films can swell, causing the resonance angles of LWs to change as a function of pH.

To optimize the response of LW to pH, precursor solutions containing 4, 5, and 6% (*w*:*v*) total monomers were used to form hydrogel films. In all cases, the molar concentration of the crosslinker was 3% of the total monomers. It was found that precursor solutions with 4% (*w*:*v*) total monomers did not always polymerize. In contrast, LWs comprising hydrogel films formed by the polymerization of precursor solutions containing 6% (*w*:*v*) monomers were less linear and were generally less responsive to pH changes than hydrogel films formed by the polymerization of precursor solutions containing 5% (*w*:*v*) monomers. Accordingly, 5% (*w*:*v*) total monomer hydrogel films were used thereafter.

A typical two-dimensional (2D) reflectivity assessment of an LW is provided in Figure 5a, where positions of TIR and resonance angles for zero (LW0)- and first (LW1)-order modes are marked. As described in Section 2.4, the 2D reflectivity curves were converted into one-dimensional (1D) reflectivity curves by drawing rectangles (shown in Figure 5a) and plotting average grayscale value versus the height of the rectangle (i.e., angle of incidence), where the grayscale values were averaged over the width of the rectangle (i.e., the distance along the width of the flow channel). Using the center of gravity algorithm, the position of dip for the resonance angle corresponding to LW0 was determined. Additionally, shifts in the resonance angle corresponding to LW0 were monitored as 10 mM phosphate buffer solutions with different pH values were introduced.

Figure 6 provides a typical plot of shifts in resonance angles versus time as 10 mM phosphate buffers with different pH values were introduced. As expected, the resonance angle of LW shifts to higher values as the pH increases from 4 to 8. This is because an LW with a hydrogel film containing DMA is expected to be more swollen and hence have a lower refractive index at pH 4 than 8. Consequentially, the resonance angle of the LW with DMA-containing hydrogel film is expected to be lower at pH 4 than at 8, which is in line with the experimental data presented in Figure 6.

As the waveguide film swells, its refractive index must asymptotically approach that of sample solution. The possible relationship between *n_w_* and *h* is given by Equation (8).(8)$$n_{w}=n_{c}+k/h$$
where *k* is a constant. We can use this model to determine the refractive index and thickness of the waveguide film at different pH values by first substituting for $n_{w}$ in Equation (5), where $n_{c}$ is known from separate refractometric measurements. The values of *k* and *h* that give the best fit to the observed changes in resonance angles can then be determined. Figure 7 shows the thickness and refractive index of a hydrogel film made using the 5% (*w*:*v*) monomer solution containing a 20% molar ratio of DMA for different pH solutions. Figure 7 shows that the hydrogel film swells by a factor of about 3, from pH 8 to pH 4, and the refractive index reduces by about 0.006 over this pH range.

Subsequently, the molar ratio of DMA in hydrogel films was varied and the shift in the resonance angle of the corresponding LWs was studied as a function of pH. Figure 8 shows that an LW with a hydrogel film containing no DMA was not responsive to pH, as exemplified by the absence of/minimal shift in resonance angle versus pH. In contrast, an LW with the hydrogel film containing 20% DMA resulted in the largest change in resonance angle, 0.3658° (0.0915° pH^−1^), over the pH range from 4 to 8. Based on the angular noise of the measurement, the resolution of the measurement is ~0.004 pH units, which compares favorably with glass pH electrodes with a resolution of ~0.01. LWs with hydrogel films containing 10% and 15% DMA were only slightly less sensitive. However, an LW with a hydrogel film containing 40% DMA showed a significantly reduced response to pH changes. Thus, LWs with hydrogel films containing 10% to 20% DMA were used for subsequent work.

### 4.2. Reversibility and Response Times

Figure 9 shows shifts in the resonance angles of an LW, comprising a hydrogel film made of a 5% (*w*:*v*) monomer solution containing a 20% molar ratio of DMA, with respect to total monomers in a cycle of alternating pH 4 and pH 8 buffers over ~167 min. As the shift in resonance angle for pH 4 and pH 8 from one cycle to another was largely the same, it can be concluded the swelling of the DMA-containing hydrogel film was largely reversible. Hence, the reported LW sensor with DMA-containing hydrogel waveguide film can be used to monitor pH for at least 5 cycles. After 5 cycles, an air bubble was introduced in the flow cell, which damaged the film. As a result, we could only study the response of the LW for up to 5 cycles.

Subsequently, we determined the response times of the LW as the buffer solution changed from pH 4 to 8 and vice versa. For this purpose, the rising and falling edges of shift in resonance angle versus time for different cycles were extracted from Figure 9. The duration of each trace was chosen such that there were sufficient leading and trailing sections to obtain a good fit, and each trace was adjusted to start at t = 0. Figure 10 clearly shows that the first cycle is somewhat different to subsequent cycles. This phenomenon has been observed by others working on pH-responsive hydrogels [38] and generally disappears after less than 10 cycles. With this hydrogel, the effect disappeared after two cycles.

Time constants for cycles of low to high and high to low pH were extracted by fitting a four-parameter sigmoid to each trace shown in Figure 10. The four-parameter sigmoid provided the time constant for each trace. Each trace represent the number and type of pH cycle. Thus, the time constants obtained from the sigmoid fits were plotted with respect to the cycle number for both types of the cycle. The resulting plot is provided in Figure 11, which shows that a transition from pH 4 to 8 (the shrinking of the hydrogel) happens more than twice as quickly as the pH 8 to 4 (swelling) transition (0.90 ± 0.14 and 2.38 ± 0.22 min, respectively) and that the time constants for the first cycle are noticeably different to those of subsequent cycles. Nevertheless, the response times of the reported LWs with DMA-containing hydrogel films were between a min to just over two mins, and hence the reported sensor is attractive for the real-time monitoring of pH.

### 4.3. Interferents

Water supplied to consumers can contain interferents such as sodium chloride, urea, aluminum sulfate, calcium chloride, and humic acid. The presence of interferents can potentially change the resonance angles of LWs because of changes in the refractive index of sample solutions, i.e., *n_c_*. Thus, the effect of the presence of interferents was studied on an LW comprising a hydrogel film made using a 5% (*w*:*v*) total monomer solution and a 10% molar ratio of DMA. The effect of sodium chloride was studied up to 2600 mg L^−1^, while the other interferents were studied up to 20 mg L^−1^. Figure 12 shows that, for concentrations of sodium chloride, urea, aluminum sulfate, calcium chloride and humic acid up to 1755, 5, 5, 20, and 2 mg L^−1^ respectively, the shift in resonance angle of the LW was minimal and hence the error in pH introduced by these interferents at up to the aforementioned levels was less than 0.1. Thus, the reported LW sensor with a waveguide film made of acrylamide and DMA, with the latter present at 10% molar ratio, is well suited to measuring pH between 4 and 8 in the presence of common interferents at typical concentration levels.

In previous work [32], we showed that stacked LWs, where one layer is pH-sensitive and the other is pH-insensitive, can be used to remove common-mode effects such as changes in the sample refractive index caused by composition or temperature changes. The synthetic hydrogel described here, with improved pH sensing characteristics compared to the chitosan hydrogel used in the previous work, could be incorporated into a fully synthetic stacked sensor.

## 5. Conclusions

Copolymer hydrogels containing a monomer with ionizable tertiary amine groups were synthesized and used for pH sensing as waveguide films in a diffraction-based leaky waveguide (LW) sensor. LWs comprising hydrogel films, made using precursor solutions with a 5% (*w*:*v*) total monomer concentration, a 20% molar ratio of pH-responsive monomer to total monomers, and a 3% molar ratio of crosslinker, gave the largest response to a change between pH 4 and 8, although 10% and 15% molar ratio hydrogel LWs were only slightly less sensitive. Modeling the change in hydrogel thickness and refractive index indicated that it swelled by a factor of about 3 between pH 8 and 4, while the refractive index reduced by 0.006. The time constants for changes between pH 4 and 8 and 8 and 4 were 0.90 ± 0.14 and 2.38 ± 0.22, respectively, over 5.5 cycles between the two pH buffers. The pH resolution was ~0.004, which compares well with other hydrogel sensors and glass pH electrodes. Finally, the reported LW sensor was found to be largely immune to common interferents, including sodium chloride, urea, aluminum sulfate, calcium chloride and humic acid, when present at typical levels.

## Figures and Tables

**Figure 1 micromachines-16-00216-f001:**
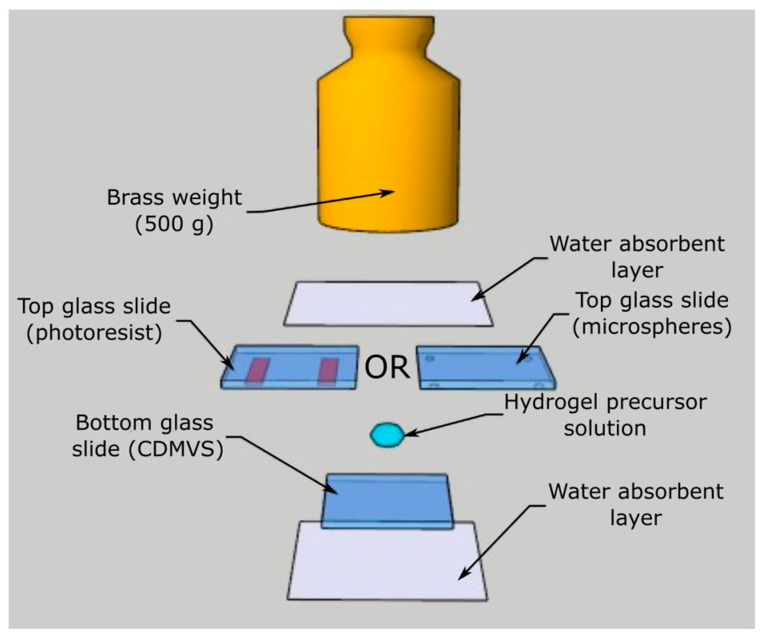
A schematic showing the casting methods used for the fabrication of LWs.

**Figure 2 micromachines-16-00216-f002:**
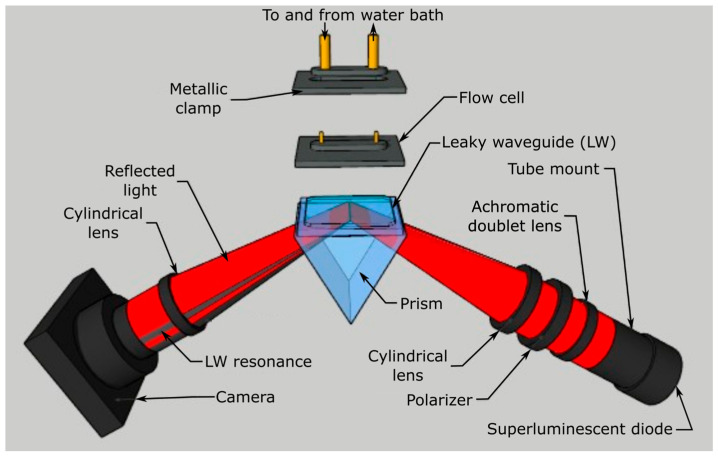
A schematic of the LW instrument.

**Figure 3 micromachines-16-00216-f003:**
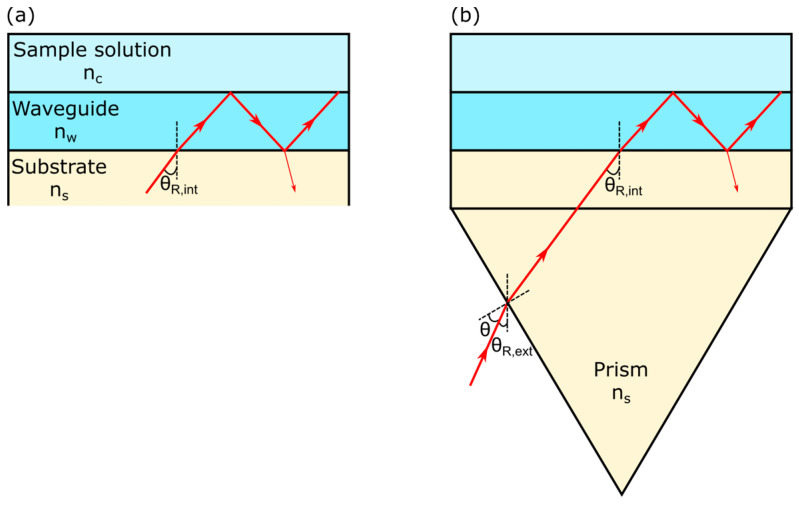
A schematic of a simplest LW (**a**) without and (**b**) with an equilateral prism where the red line with arrows show the path of light beam.

**Figure 4 micromachines-16-00216-f004:**
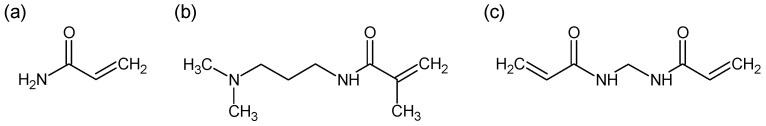
Chemical structures of (**a**) acrylamide, (**b**) N-[3-(dimethylamino)propyl]methacrylamide (DMA) monomers, and (**c**) N, N-methylene bisacrylamide crosslinker.

**Figure 5 micromachines-16-00216-f005:**
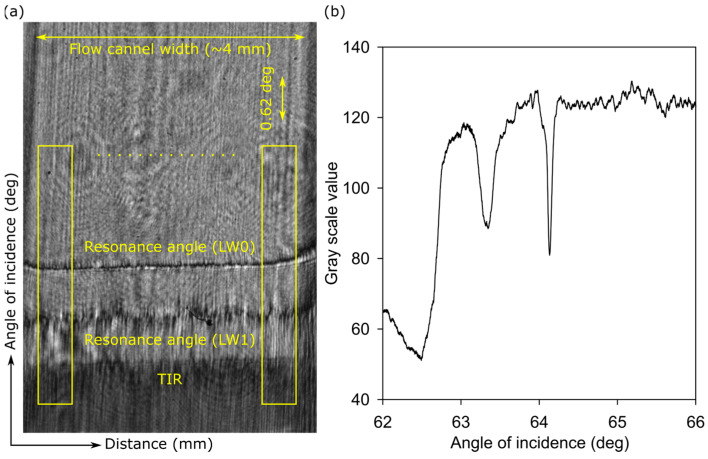
A typical (**a**) two-dimensional reflectivity curve and the corresponding (**b**) one-dimensional reflectivity curve of an LW with a hydrogel film made of 5% (*w*:*v*) total monomer solution and with a 15% molar ratio of DMA to total monomer.

**Figure 6 micromachines-16-00216-f006:**
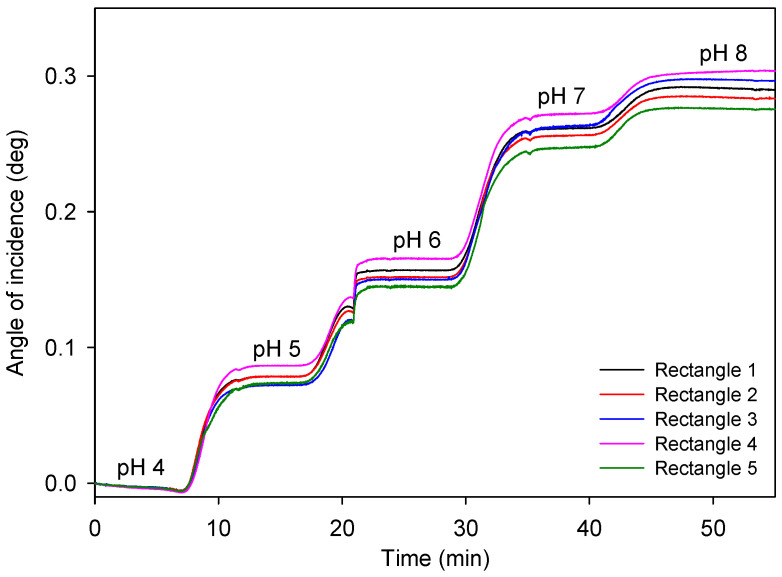
Time course of the response of an LW to pH where the hydrogel film was made of 5% (*w*:*v*) total monomer solution and 15% molar ratio of DMA to total monomer.

**Figure 7 micromachines-16-00216-f007:**
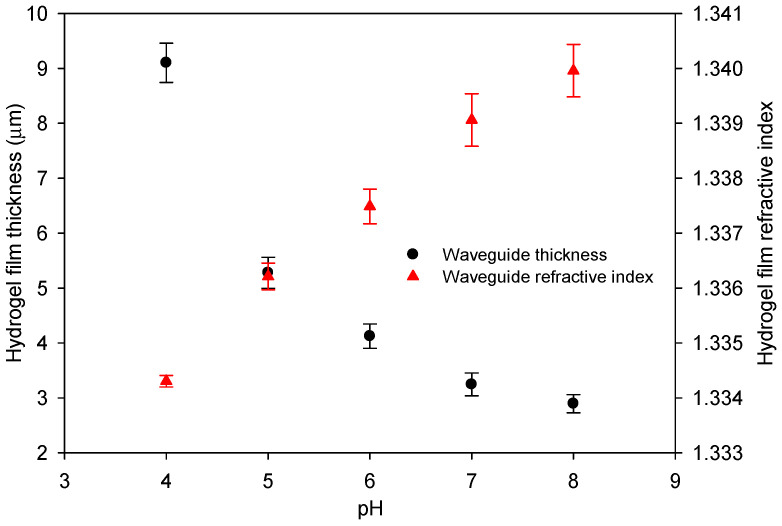
Estimated thickness and refractive index of waveguide films made of 5% (*w*:*v*) monomer solution containing a 20% molar ratio of DMA for different pH solutions.

**Figure 8 micromachines-16-00216-f008:**
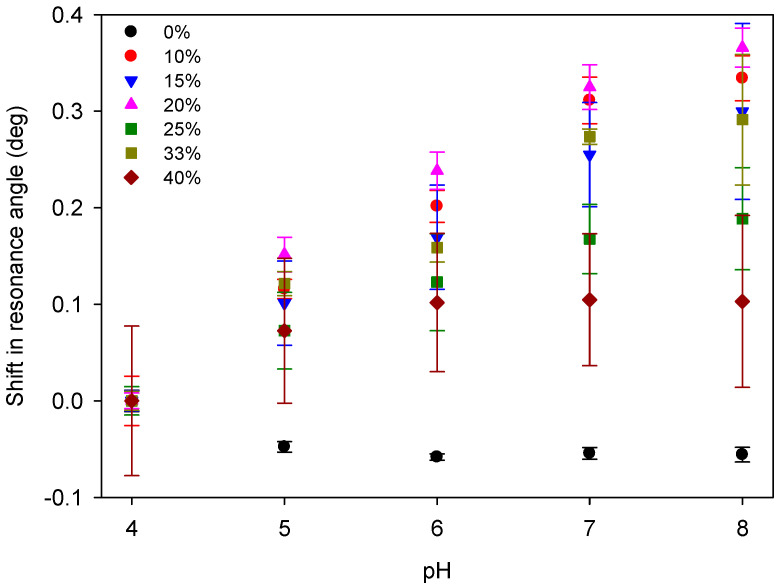
The response of an LW comprising a 5% (*w*:*v*) hydrogel film to pH, where the molar ratio of DMA to total monomer was between 0 and 40% (where error bars show variations in response of LWs across the width of the flow cell).

**Figure 9 micromachines-16-00216-f009:**
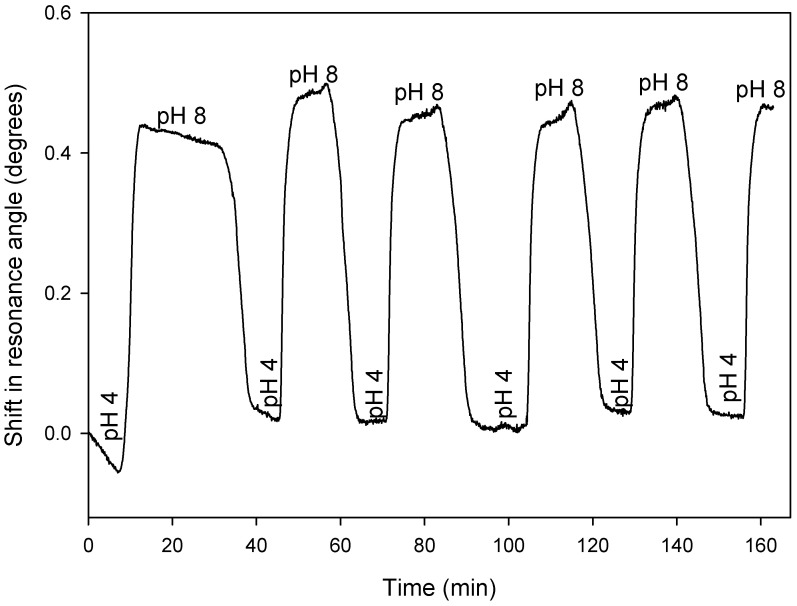
The response of an LW, comprising a hydrogel film made of a 5% (*w*:*v*) monomer solution containing a 20% molar ratio of DMA, to cycles of pH 4 and 8 buffers.

**Figure 10 micromachines-16-00216-f010:**
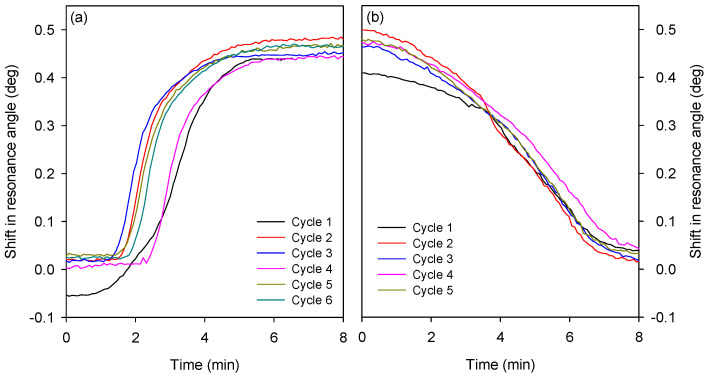
Shift in resonance angle versus time of an LW exposed to cycles of (**a**) pH 4 to pH 8 and (**b**) pH 8 to pH 4.

**Figure 11 micromachines-16-00216-f011:**
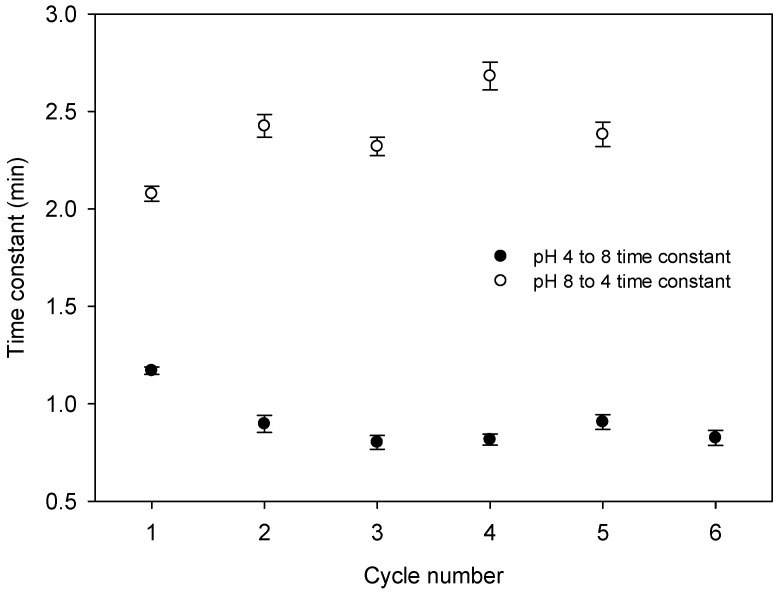
Time constant of an LW exposed to cycles of pH 4 to pH 8 and pH 8 to pH 4.

**Figure 12 micromachines-16-00216-f012:**
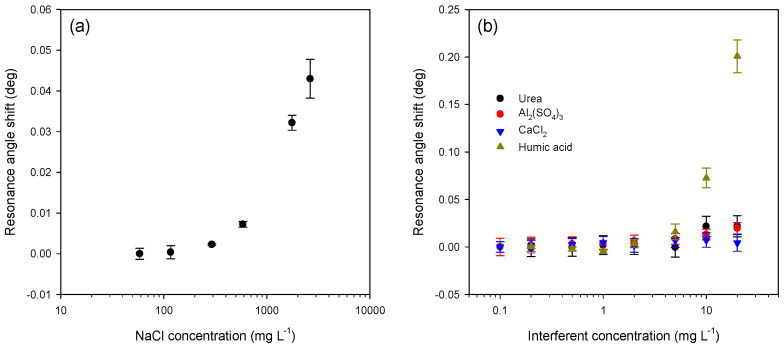
Shits in resonance angles of an LW comprising a hydrogel film made using a 5% (*w*:*v*) total monomer solution and a 10% molar ratio of DMA for 10 mM phosphate a buffer of pH 7 containing (**a**) sodium chloride, (**b**) urea, aluminum sulfate, calcium chloride, and humic acid.

## Data Availability

The original contributions presented in this study are included in the article. Further inquiries can be directed to the corresponding author.

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
