# Peer review of "pH Measurements Using Leaky Waveguides with Synthetic Hydrogel Films"

_micromachines, 2025, doi:10.3390/mi16020216_

Round 1
Reviewer 1 Report
Comments and Suggestions for Authors
Recommendation: Major revision.
Manuscript ID: micromachines-3422243
Comments:
This manuscript introduces a pH sensor based on leaky waveguides (LWs) utilizing synthetic hydrogel films comprising acrylamide and N-[3-(dimethylamino)propyl] methacrylamide (DMA) as monomers. The LW sensor offers advantages such as simplicity, insensitivity to typical interferents, and reversible operation. The authors report a pH sensitivity range of 4–8, with response times of 0.90 ± 0.14 minutes (pH increase) and 2.38 ± 0.22 minutes (pH decrease). The work emphasizes the benefits of synthetic hydrogels over natural materials, which are prone to variability. The hydrogels demonstrated excellent linearity and reversibility for pH sensing over multiple cycles. Furthermore, the sensor's performance remained robust in the presence of interferents like sodium chloride, urea, and humic acid. The manuscript provides detailed descriptions of sensor fabrication, operational principles, and analytical validation, presenting a promising solution for real-time pH monitoring. Although the topic is very interesting, this manuscript contains many ambiguous points. Detailed comments are listed below.
1. The introduction provides adequate background; however, consider elaborating on the practical implications of the sensor in specific applications like environmental monitoring or healthcare.
2. Could the authors highlight how this LW sensor differs from or improves upon previous synthetic hydrogel-based pH sensors?
3. The fabrication process is well-documented. Would providing more quantitative data on the reproducibility of the casting process (e.g., hydrogel film thickness variations) strengthen the methodology section?
4. Can the authors clarify the rationale for using specific DMA concentrations (10–20%) in the hydrogels?
5. In Figure 11, the response times for the first cycle differed from subsequent cycles. Could the authors explore this observation further or provide mechanistic insights into the initial conditioning effects?
6. The resonance angle measurements seem critical. Can the authors discuss the resolution and accuracy of the detection system, especially under different interferent conditions?
7. How does the LW sensor compare with conventional glass electrodes in terms of sensitivity and long-term stability?
8. Given that temperature variations affect hydrogel swelling, has the sensor been evaluated under varying temperature conditions?
9. The reported pH range is 4-8. Are there potential modifications to expand this range?
10. The format of the reference needs to be in accordance with the format required by Micromachines. There are a lot of format errors in the manuscript, for example: reference 19.
11. The format of figure caption should be consistent. For example: “Figure 1. A schematic showing the casting methods used for fabrication of LWs” should be “Figure 1. A schematic showing the casting methods used for fabrication of LWs”.
Reviewer 2 Report
Comments and Suggestions for Authors
The manuscript developed a pH sensor utilizing leaky waveguides (LW) with synthetic hydrogel films, intended for pH measurements in water supply systems. The manuscript initially introduced optical methods for pH measurements. Subsequently, it presented the chemicals, materials, and manufacturing processes involved in the creation of LWs, along with a description of the LW instrument. The working principle of the sensor was extensively explained using mathematical equations. Experimental investigations were conducted on the pH response of the sensor, including analysis of LW reflectivity curves, the time response of LWs to various pH values, changes in estimated thickness and refractive index with different pH solutions, and shifts in the resonance angle associated with varying densities of the hydrogel film. Measurements were also taken for reversibility, response time, and interference. However, there are several issues that need to be addressed:
1. In the abstract section, various optical methods for pH measurement were introduced. However, a more comprehensive comparison between pH sensors is necessary, including additional information on parameters such as pH value range, accuracy, and sensitivity if possible.
2. In figure 1, there are two top glass slides. An "or" indicater is necessary between them to prevent confusion.
3. In Section 3.1, the working principle should be presented in the "Principle" section before the "Results" section.
4. In Figure 9, the total time on the horizontal axis is approximately 160 seconds. However, in the text it states that it was "performed over ~167 min." Please confirm if this unit is seconds or minutes.
5. In Figure 9, there is least 5 cycles, what factors determine number of the cycles?
6. How were Figure 10 and 11 obtained? There are no decription in the text.
Round 2
Reviewer 1 Report
Comments and Suggestions for Authors
The manuscript can be accepted in the current state.